# Genome Expression Dynamics Reveal the Parasitism Regulatory Landscape of the Root-Knot Nematode *Meloidogyne incognita* and a Promoter Motif Associated with Effector Genes

**DOI:** 10.3390/genes12050771

**Published:** 2021-05-18

**Authors:** Martine Da Rocha, Caroline Bournaud, Julie Dazenière, Peter Thorpe, Marc Bailly-Bechet, Clément Pellegrin, Arthur Péré, Priscila Grynberg, Laetitia Perfus-Barbeoch, Sebastian Eves-van den Akker, Etienne G. J. Danchin

**Affiliations:** 1INRAE, Institut Sophia Agrobiotech, Université Côte d’Azur, CNRS, 06903 Sophia Antipolis, France; martine.da-rocha@inrae.fr (M.D.R.); julie.dazeniere@gmail.com (J.D.); marc.bailly-bechet@inrae.fr (M.B.-B.); arthur.pere@inrae.fr (A.P.); laetitia.zurletto@inrae.fr (L.P.-B.); 2Department of Plant Sciences, University of Cambridge, Cambridge CB2 3EA, UK; cb2122@cam.ac.uk (C.B.); cp678@cam.ac.uk (C.P.); 3School of Medicine, Medical & Biological Sciences, University of St. Andrews, St Andrews KY16 9AJ, UK; pjt6@st-andrews.ac.uk; 4Embrapa Genetic Resources and Biotechnology, Brasília 70770-917, DF, Brazil; priscila.grynberg@embrapa.br

**Keywords:** genome, transcriptome, phytoparasitism, gene expression regulation, *Meloidogyne*, nematode

## Abstract

Root-knot nematodes (genus *Meloidogyne*) are the major contributor to crop losses caused by nematodes. These nematodes secrete effector proteins into the plant, derived from two sets of pharyngeal gland cells, to manipulate host physiology and immunity. Successful completion of the life cycle, involving successive molts from egg to adult, covers morphologically and functionally distinct stages and will require precise control of gene expression, including effector genes. The details of how root-knot nematodes regulate transcription remain sparse. Here, we report a life stage-specific transcriptome of *Meloidogyne incognita*. Combined with an available annotated genome, we explore the spatio-temporal regulation of gene expression. We reveal gene expression clusters and predicted functions that accompany the major developmental transitions. Focusing on effectors, we identify a putative cis-regulatory motif associated with expression in the dorsal glands, providing an insight into effector regulation. We combine the presence of this motif with several other criteria to predict a novel set of putative dorsal gland effectors. Finally, we show this motif, and thereby its utility, is broadly conserved across the *Meloidogyne* genus, and we name it Mel-DOG. Taken together, we provide the first genome-wide analysis of spatio-temporal gene expression in a root-knot nematode and identify a new set of candidate effector genes that will guide future functional analyses.

## 1. Introduction

Plant parasitic nematodes are microscopic worms that threaten the security of most major agricultural crops such as soybean, cotton, peanut, banana, coffee, and potato [1,2]. Damage caused by nematodes therefore represents an important constraint on global food security in the developed and developing world and is estimated to cost world agriculture 173 billion US dollar per year [3]. The majority of these loses are attributed to the obligate biotrophic sedentary endoparasites: the root-knot nematodes (RKN; *Meloidogyne* spp.) and the cyst nematodes (CN; *Heterodera* spp. and *Globodera* spp.).

Root-knot nematodes are considered obligate biotrophic endoparasites, as they spend the majority of their life cycle within host roots and feed from living tissues. The typical RKN life cycle is complex and usually completed in 4–6 weeks, although it is heavily dependent on the host plant and nematode species, as well as environmental conditions such as temperature and moisture. Regardless of the timing, a number of key transitions are required for successful completion of the life cycle and consequently of plant parasitism [4]. Generally, ~500 eggs are deposited in a glycoproteic gelatinous matrix, secreted by the rectal glands of adult females at the root surface, to protect against biotic and abiotic stress [5]. Within the eggs, embryonic development leads to a first stage, vermiform and coiled juvenile (J1), which then molts into a second stage juvenile (J2). Favorable moisture and temperature conditions trigger hatching of vermiform J2s from the egg, which constitutes the only infective stage that is able to penetrate root tissue, and is thus also called pre-parasitic J2 (ppJ2). Through the perception of root diffusates and other factors, the ppJ2 migrates towards the host root system to subsequently enter the plant tissues, often close to a root tip. Root penetration is made possible by a combination of mechanical and enzymatic disruption of the plant cell wall. At this stage, cell wall degrading/modifying enzymes (referred to as CWMEs) are secreted through a syringe-like stylet connected to secretory gland cells. Once inside the plant, the nematode migrates intracellularly between the cortical cells towards the root tip [6]. Upon reaching the host vasculature, the parasitic nematode induces the formation of five to seven hypertrophied and multinucleated giant cells from parenchymal root cells via the secretion of effector proteins and other molecules that manipulate host cell division and other functions. These metabolically hyperactive giant cells will remain the sole source of nutrition for the developing endoparasite. Once the permanent feeding site is initiated, the nematode uptakes plant nutrients through its stylet and rapidly becomes a swollen fusiform parasitic J2. In a compatible interaction, the parasitic J2 undergoes two successive molts to reach the J3 and then J4 juvenile stages. These two stages are characterized by the absence of a functional stylet and hence the nematode does not feed at these stages [5,7]. Late J2, J3, and J4 stages are morphologically difficult to differentiate, and this can only be achieved by a meticulous examination to identify the presence of a functional stylet or to count the number of superimposed outer cuticles resulting from the successive molts [7]. The last transition leads to adult individuals, with a marked dimorphism between males and females. Males return to a vermiform morphology, and no evidence for feeding at this stage has been shown. Males are extremely rare and do not participate in the generation of offspring in obligate parthenogenetic *Meloidogyne* species such as *M. incognita*. The proportion of males can be higher in conditions unfavorable to the development of adult females and production of offspring. Adult females are morphologically distinct; being pear-shaped and having a functional stylet, they resume feeding and produce hundreds of eggs that will eventually form the next generation of parasitic juveniles and thus complete the life cycle. The parasitic life cycle of root-knot nematodes is characterized by successive transitions punctuated by molts and profound morphological and functional changes.

Accompanying these morphological changes, successful manipulation of host physiology, structure, development, and immunity must be achieved. It is generally accepted that this is achieved by the delivery of a repertoire of effectors into and around host cells. Effectors in plant-parasitic nematodes are primarily produced in two sets of pharyngeal glands: dorsal gland (DG) and subventral gland (SvG) cells [8,9]. The SvGs are primarily, although not exclusively, active at the earlier stages of parasitism, and are therefore thought to contribute to J2 penetration and migration. The DGs are primarily, although not exclusively, active at the later stages of infection, and are therefore thought to contribute to the development and maintenance of the feeding site [10]. Hence, PPNs must control the transcription of effectors and other genes in space and time to successfully parasitize their host, while undergoing profound morphological changes of their own.

Advances in genomics and transcriptomics provide a means to explore the nature and regulation of parasitic nematode genomes during infection. Notably, large scale transcriptome studies of various different plant-parasitic nematode groups have revealed global trends in gene expression at different stages of the life cycle [11], and important insights in the transcriptional regulation of effectors. An emerging theme in many different plant-parasitic nematodes is the presence of conserved putative cis regulatory elements that are associated with effectors expressed in specific gland cells [12]. For example, the DOG box is conserved in cyst nematodes [11,13], enriched in the promoters of DG effectors [11,13,14], and can be used to predict novel effectors [11,13,14,15]. These analyses have also been expanded to the migratory endoparasites (e.g., *Bursaphelenchus xylophilus* STATAWAARS motif [14]).

Even though RKNs are the major contributors to crop losses due to nematodes, no study to date has comprehensively described the transcriptional changes across their life cycle, nor explored the cis-regulatory elements associated with effector expression in this group. Here, we report a comprehensive transcriptomic survey of *M. incognita* using life stage-specific RNA-seq data to understand the global expression changes and to profile parasitism genes throughout a complete pathogen reproductive cycle. Availability of an annotated genome sequence comprehensively representing the set of protein-coding genes [16] coupled with this RNA-seq time course provides an opportunity to explore the spatio-temporal regulation of gene expression in *M. incognita*, with a focus on effectors. Our study reveals gene expression variations that accompany the major developmental transitions. We identify the first putative cis-regulatory motif associated with tissue specific expression in root knot nematodes, providing a novel insight into effector regulation in the DG cell. We combine the presence of this motif with several other criteria to predict a novel set of putative DG effectors in this species. Finally, we show this motif, and thereby its utility, is broadly conserved across the *Meloidogyne* genus, and we name it Mel-DOG.

## 2. Materials and Methods

### 2.1. Gene Expression Levels during Four Developmental Life Stages of M. incognita

To determine the expression levels of protein-coding genes, we used RNA-seq data of four life stages of *M. incognita*, strain Morelos (ppJ2s, parasitic J3/J4, adult females, and eggs) produced in triplicate in a previous analysis [16] and that have been recently re-mapped to the genome with STAR [17] using more stringent end-to-end parameters [18].

Read counts and expected read counts were calculated on the predicted genes from the *M. incognita* GFF annotation [16] as FPKM and TPM values using two different methods:With RSEM [19] to take into account the multi-mapped reads via expectation maximization as explained in [18]; andWith htseq-count [20], where only uniquely mapped reads are counted.

### 2.2. Identification of Differentially Expressed Genes between Life Stages

We used three independent methods to identify genes differentially expressed between the four life stages. RSEM expected read counts were used to identify differential gene expression using EBseq [21]. Read counts from htseq count were used to identify differential gene expression using DEseq2 [22] and EdgeR exact tests [23]. We considered genes to be differentially expressed if they returned a log2 fold change value > 2 and a false discovery rate (adjusted *p*-value) < 0.05, consistently, from all of the three above-mentioned methods.

### 2.3. Clustering of Differentially Expressed Genes

To classify differentially expressed genes in expression clusters, we used the log10 of one plus the median FPKM value over the 3 replicates as determined by RSEM and as previously published in [18] with data publicly available at https://doi.org/10.15454/YM2DHE (accessed on 13 May 2020). For gene expression data, the goal of a clustering approach is to minimize the divergence within a cluster yet maximize the divergence between clusters. We used a distance based on the Pearson correlation (*r*) to estimate this divergence between gene expression patterns. This metric is appropriate for temporal variations of expression values such as a developmental life cycle. Many differential gene expression analyses employ an arbitrarily predetermined number of clusters in which genes have to be assigned according to their expression pattern. This kind of approach is not ideal but determining an optimal number of clusters is far from evident. Here, because our goal was to obtain a low number of clusters, we used a divisive clustering.

Divisive approaches first divide the dataset into the two most dissimilar clusters and progress until the desired number of clusters is reached. For the divisive analysis, we used the Diana method in the cluster package of R. To determine the most appropriate number of clusters, we used the ClusterSim packages in R. We used three indicator metrics (CH, Silhouette and C-index). We allowed the number of clusters to vary from a minimum of 2 to a maximum of 25.

### 2.4. Identification of Overrepresented Gene Ontology Terms

Gene Ontology (GO) terms have been assigned to the *M. incognita* protein-coding genes based using InterProScan [24] in a previous analysis [18], with all the results publicly available at https://doi.org/10.15454/9BFFKG (accessed on 13 May 2020).

To identify overrepresented GO terms, we used a hypergeometric test as implemented in func [25] within the R package GOfuncR. Overrepresented GO terms were searched at each transition between the four life stages and within each of the expression clusters. A family-wise error rate (FWER) threshold value of 0.05 divided by the number of comparisons was set to consider a GO term as significantly overrepresented in the dataset.

### 2.5. Mapping of Genes Known to Be Specifically Expressed in Secretory Gland Cells on Meloidogyne Genomes

Based on the literature, we established a manually curated list of *M. incognita* genes that are specifically expressed in secretory sub-ventral gland (SvG) or dorsal gland (DG) cells. We started from the list published in [26] and manually checked the figures associated with the cited references to only keep genes with evidence for specific expression via in situ hybridization. We completed this list with the Mi-PNF3 gene, published more recently [27], and eliminated redundancy between the sequences at the protein level. We retrieved the corresponding CDS sequences and aligned them to the *M. incognita* genome sequence [16] using the splice aware aligner SPALN [28] with the following parameters:

‘spaln -Q7 -O4 -M -d Mi.mfa test-CDS > result.gff

-M = report multiple hits and not only the best

-Q4 = DP mode, 5–7 = maximum number of reported hsp

-O4 = megablast-like output format

To investigate the conservation of effectors in the *Meloidogyne* genus, we also aligned these same CDS to the genomes of the evolutionary close relatives *M. arenaria, M. javanica,* and *M. enterolobii* [16,29], as well as the more distant *M. hapla* [30], using the same parameters.

### 2.6. Identification of Specific Motifs in the Upstream Regions of Genes Expressed in Secretory Gland Cells

To identify putative regulatory motifs associated with effector genes, a custom Python program was written, with unit tests, to extract intergenic regions in the *M. incognita* genome (https://github.com/peterthorpe5/intergenic_regions, accessed on 13 May 2021). Various length regions (up to 10,000 bp) 5′ of the coding start site (as defined in the gff3 annotations) were extracted from the genome for all genes. If another gene was located within this region, the script returned the sequence between the two coding regions. The resulting intergenic regions were divided into positive and negative test sets, and taken forward for further motif analysis.

As a positive test set, we retained *M. incognita* genomic loci corresponding to alignments of the known SvG and DG genes as described above, encompassing the start (ATG) position and covering at least ⅔ of the effector length with a minimum of 99% identity. When several effectors mapped onto an overlapping locus, we kept the best scoring alignment. When one effector mapped to multiple loci, we kept up to three of the best-scoring alignments matching the above-mentioned criteria.

As a negative set, we selected 167 *M. incognita* genes from an Orthofinder [31] analysis previously performed on 64 genomes, including 62 nematodes and two outgroup tardigrade species [18]. The selection criteria were as follows. (i) The vast majority (163 genes) came from orthogroups conserved in at least 80% of the species and 85% of the Tylenchida and did not form multigene families. Being widely conserved across nematodes, including in many non-phytoparasitic species, and not forming multigene families, these genes are more likely involved in core housekeeping functions than coding for effector proteins. (ii) Four genes specific to root-knot nematodes but bearing no signal peptide for secretion, including one gene specific to *M. incognita*. The annotated list was deposited at https://doi.org/10.15454/KYPEN0.

Promoter regions (n bp upstream of the start codon) were analyzed for the presence of enriched motifs as previously described [11], with the following modifications. Enrichment of motifs between categories (DG versus non-effectors, DG versus SvG, etc.) was calculated using HOMER [32] with default parameters. Instances of the motif were identified in FASTA sequences of promoter regions using the FIMO web server [33]. Correlations between the number of motifs in a given promoter and the presence or absence of signal peptides and transmembrane domains encoded by the corresponding gene were calculated. To identify variants of the motif that are able to similarly predict whether the corresponding gene encodes a putatively secreted protein, a custom Python script was written. The script identifies the occurrence of all 1 bp mismatch variants of the motif in the promoter regions of all genes and calculates the proportion of genes that encode putatively secreted proteins for 1, 2, or 3 copies of each motif variant (https://github.com/sebastianevda/Mel_DOG_scripts, accessed on 13 May 2021). The same initial motif identification procedure (HOMER) was repeated on the genomes of the four other *Meloidogyne* species introduced in the above section using 2 kb promoter regions.

### 2.7. Identification of Putative Secreted Proteins and Effectors

We used the list of 2811 *M. incognita* proteins bearing a signal peptide for secretion, and no transmembrane region, as putative secreted proteins (PSP) as described in [18]. The list is publicly available at https://doi.org/10.15454/JCYZDI (accessed on 13 May 2020). To further differentiate candidate effectors from the rest of the PSP, we only retained PSP bearing at least one of the four motifs enriched in known *M. incognita* effectors in their first 100 amino acids, as explained in [18,34]. The list of 2146 PSP bearing an effector protein motif is publicly available at https://doi.org/10.15454/CSTXU2 (accessed on 13 May 2020).

### 2.8. Evolutionary Origin of M. incognita Effectors

Based on the mapping of non-redundant known *M. incognita* SvG and DG effectors on the genomes of five *Meloidogyne* species, ancestral numbers were deduced across their phylogeny using parsimony inference with Mesquite [35]. In the genomes of *M. incognita*, *M. arenaria*, *M. javanica,* and *M. enterolobii*, only genes mapping with >90% identity on >66% of their length were considered unambiguously present (character state ‘1′ in Mesquite). Genes with significant SPALN mapping but that did not meet the % identity and % coverage thresholds were considered as unsure to be present (character state ‘?’ in Mesquite). Given that *M. hapla* is phylogenetically much more distant, only a length coverage threshold was used to infer unambiguous presence following a significant SPALN mapping from the ATG position. Genes that returned no significant SPALN mapping were considered absent from the genomes under consideration (character state ‘0′ in Mesquite). The tree topology of the five *Meloidogyne* species used for ancestral states reconstruction was taken from [36]. We used the Mesquite analysis ‘Trace all Characters’ to reconstruct characters (presence/absence of each effector) at each ancestral branch. To infer the minimal unambiguous number of effectors at each ancestral branch, the sum of 1′s at this branch was calculated, while the equi-parsimonious 0/1′s ancestral states were considered an absence in a conservative scenario. All the data used to produce the ancestral reconstruction, as well as the raw results, were deposited at https://doi.org/10.15454/OJMRDD.

## 3. Results

### 3.1. Predicted Functions of Differentially-Expressed Genes Are Consistent with Transitions between Developmental Stages

During the course of the *M. incognita* life cycle, the nematode undergoes several major transitions, from a sedentary egg to a migratory second stage juvenile in the soil, culminating in a sedentary endoparasite within the living host root system. To understand the changes in gene expression during these transitions, and progression through the *M. incognita* life cycle in general, the present study deployed RNA sequencing (RNA-seq) of four time points: eggs, ppJ2s, a mixture of larval J3/J4s, and adult females [16]. Using these RNA-seq data, differential gene expression was analyzed to capture biological signatures between these different developmental transitions.

Approximately 30% of the protein-coding genes (12,461) are significantly differentially expressed between at least two developmental life stages, according to three independent statistical methods (https://doi.org/10.15454/VLN8UC). Interestingly, all three methods converge on similar log2 (fold change) values, despite using different counting strategies. Three of the four transition stages are rich (>4500) in differentially expressed genes (DEGs), with the exception of the transition from larval J3/J4 to adult females, with the smallest number of DE genes (555), which was nearly 10-fold lower (Figure 1A and https://doi.org/10.15454/VLN8UC). The largest number of DEGs (6967) was observed in the transition between the pre-parasitic J2s and the mix of J3/J4 sedentary stages. Interestingly, this transition is generally characterized by large scale gene repression: the vast majority of DEGs (5196) were significantly downregulated (log2FC down to <−16) and only 1771 were significantly upregulated (log2FC up to >14). Moreover, the largest upregulation in gene expression (3948 DEG at log2FC up to 10.9) was observed at the developmental transition from adult females to eggs. A total of 9145 DEGs (70%) were differentially expressed between multiple comparisons, while 3271 were differentially expressed specifically at one given transition. The highest number of specifically differentially expressed genes (607) was from pre-parasitic J2 to mixed J3/J4. Together, our filtered dataset provides an initial comprehensive view of transcription during the entire *M. incognita* life cycle.

To gain insight into the biological changes occurring during the time-course of the *M. incognita* life cycle, we identified and analyzed significantly overrepresented GO terms in genes differentially regulated between each pair of adjacent life stages (methods, Figure 1, and https://doi.org/10.15454/VLN8UC). At the transition from eggs to ppJ2s, a large proportion of upregulated DEGs have GO terms related to sensory perception (e.g., cell communication, GPCR signaling, neuropeptide signaling, response to stimulus etc.; https://doi.org/10.15454/NOT2LH). The GO terms related to locomotion/movement (e.g., myofibrils, contractile fibers, striated muscle thin filament, etc.) were also enriched. Significantly enriched GO terms in this transition were also related to carbohydrate metabolism, including those related to plant cell wall degradation (e.g., carbohydrate degradation, hydrolases active on glycosyl bonds, polygalacturonase activity, pectate lyase activity, etc.). The parasitic J2s are known to secrete a range of carbohydrate-active enzymes (CAZymes, including cellulases, pectate lyases, and polygalacturonases), many of which were acquired via horizontal gene transfer, for the degradation of the plant cell wall to facilitate penetration and movement, and also to supply nutrients to the animal [37,38]. Another notable feature was the enrichment of GO terms including iron binding, heme binding, and oxidoreductase activity, which may indicate the beginning of a response to oxidative stress at this transition. The cellular component GO term ‘extracellular region’ was also overrepresented in this transition, consistent with the secretion of many enzymes and other proteins at this stage.

At the transition from ppJ2s to sedentary larval J3/J4s, terms related to biotrophic parasitism were overrepresented. GO terms related to response to stressors were upregulated (e.g., response to oxidative stress, oxidation–reduction processes, oxidoreductase activity, peroxidase, antioxidant; https://doi.org/10.15454/VE2MLH). Other overrepresented terms were related to host colonization and/or survival, such as serine-type peptidase activity and exopeptidase activity. Furthermore, we noted a downregulation of genes coding for sensory perception (i.e., G protein-coupled receptor signaling pathway, neuropeptide signaling pathway, chemical synaptic transmission), indicating that the perception of environmental cues may be less important once the nematode has established its feeding site. Other interesting enriched GO terms in upregulated genes encompassed the following main functional categories related to lipid biosynthesis, transport, and storage (e.g., fatty acid synthase, lipid transporter activity, lipid transport, lipid localization), reflecting a need to store nutrients and energy to sustain the important metamorphosis to the next stage (adult). As these transitions involve metamorphoses and important morphological/structural changes, it is also consistent with other overrepresented terms such as structural constituents of the cuticle and chitin binding. This transition reveals a marked shift in the nematode lifecycle, with the differential regulation of various genes promoting growth and pathogenic success.

At the transition from sedentary J3/J4 larvae to adult females, the nematode resumes its feeding phase and develops hundreds of eggs. We observed that genes coding for locomotion-related functions were downregulated (i.e., the regulation of muscle contraction), confirming that most nematode movements have stopped. The overrepresented GO terms associated with upregulated genes (https://doi.org/10.15454/5UJNHS) were similar to the ones identified at the previous transition (from ppJ2 to sedentary J3−J4 stages). These GO terms included those related to lipid transport and localization (e.g., lipid transport, lipid localization, lipid transporter activity, macromolecule localization) and related to defense against oxidative stress (e.g., response to oxidative stress), or possible remodeling of the cuticle (e.g., chitin metabolic processes, chitin binding). These GO terms support the fact that the processes and functions triggered at the previous transition are further amplified at this transition.

By contrast, comparing the eggs to the adult female stages (https://doi.org/10.15454/6EYGIX) showed that most of the induced distinctive DEGs were associated with biological processes (such as ion transport, ion transmembrane transport, metal/potassium ion transport), related to DNA dynamics (i.e., DNA metabolic process, DNA replication, DNA integration), and also involved in movement of the cell. Important changes in gene expression were observed with overrepresented GO terms associated with molecular function, such as ion channel activity, channel activity, transmitter-gated ion channel activity, potassium channel activity, calcium/metal ion binding, DNA-directed DNA polymerase, DNA binding. In the cellular component category, there were a significantly high number of DEGs classified as plasma membrane, membrane, synaptic membrane, cation channel complex. The distribution of DEGs may be correlated with an overall transcriptional change in gene expression related to embryonic development. In addition, we observed a significant downregulation of genes encoding stress responses (i.e., superoxide dismutase, oxido-reductase activity, L-ascorbic acid binding), digestion enzymes (serine-type endopeptidase inhibitor activity, metalloendopeptidase inhibitor activity, hydrolase activity) and general metabolism (i.e., carbohydrate binding, lipid transporter activity, galactosyltransferase activity), underlying that the nematode has stopped feeding at this stage and by extension completes its parasitic lifecycle.

### 3.2. Most of the Known M. incognita Effectors Are Conserved in Multiple Meloidogyne Genomes and Were Probably Inherited from a Common Ancestor

To explore the transcriptional regulation of parasitism, we first identified genome loci that encode known effectors with experimental evidence of expression in the two sets of pharyngeal glands: sub-ventral (SvG) or dorsal pharyngeal gland (DG) cells. From the literature, we identified 48 SvG and 34 DG non-redundant *M. incognita* effector genes and studied their conservation in the *M. incognita* and other root-knot nematode genomes (https://doi.org/10.15454/P5YIGX). Two anomalies were noted. Firstly, the effector 35A02 (msp25), described as specifically expressed in the SvG in [39], is 97% identical to Minc02097, conversely detected in the DG in [40]. Both genes map to the same genomic location in *M. incognita* and to the same CDS (Minc3s00202g07465). Secondly, Minc18033, described as SvG-specific in [40], is 99% identical to 16E05 (=msp17), which is described as DG-specific in [39]. Both Minc18033 and 16E05 map the same genomic locations and the same two CDSs (Minc3s02105g28312 and Minc3s03024g32468). In both cases, we eliminated redundancy between the two sets, and assigned the most recent traceable gland cell location.

All of the 48 known and non-redundant SvG effector genes did mapfrom the ATG position to the *M. incognita* genome assembly (https://doi.org/10.15454/P5YIGX). These gene candidates mapped to 80 different loci in the genome, which is consistent with the allopolyploid genome structure [16]. A total of 45 SvG effector CDS mapped with at least 90% identity and across >66% of their length; the three other effectors, 31H06, Mi-Pel1, and Minc03866, did not meet the 90% identity threshold (% identities ranged between 85–89%). The 45 SvG effectors mapping with high confidence corresponded to 74 different loci and 64 predicted gene models in the *M. incognita* annotated reference genome (https://doi.org/10.15454/P5YIGX).

The 48 non-redundant *M. incognita* SvG effectors were mapped to the genomes of three other polyploid and parthenogenetic species (*M. arenaria*, *M. javanica,* and *M. enterolobii*) belonging to the same *Meloidogyne* clade I than *M. incognita* [41] to determine their evolutionary conservation. All (48/48) mapped from the ATG with >90% identity and across >66% of their length to at least one of the above-mentioned genomes. Most of these effectors mapped to multiple loci in each of these genomes, which is again consistent with their polyploid structures. Only eight, five, and two effectors did not meet the 90% identity and 66% coverage thresholds in the genomes of *M. arenaria*, *M. javanica* and *M. enterolobii*, respectively (of which three, one, and none did not map at all, respectively). The reconstruction of ancestral states from the phylogenetic distribution of SvG effector genes across these *Meloidogyne* species indicated that at least 42/48 effectors were present in a common ancestor of clade I root-knot nematodes (Figure 2). To assess whether these SvG effector genes were evolutionarily more anciently conserved, we also mapped them to the genome of *M. hapla*, a facultative parthenogenetic and diploid species belonging to the more distantly related clade II [41]. Only 25/48 effectors mapped from the ATG on at least 66% of their length. However, due to the higher evolutionary distance of *M. hapla*, no alignment reached 90% identity (range = 63.8–89.2%). Except Mi-VAP2, all the mapping candidate effectors matched one single locus in the *M. hapla* genome, consistent with its diploid homozygous structure. Overall, at least 52% (25/48) of the SvG effectors present in *M. incognita* were probably inherited from a common ancestor of clade I and clade II Meloidogyne, the rest either being actually absent from that ancestor, too distantly diverged to map on the *M. hapla* genome, or actually missing from this genome assembly (Figure 2).

Of the 34 non redundant effector genes with DG-specific expression, 33 did map from the ATG position to the *M. incognita* genome assembly. One known DG effector, 28B04, did not map on the *M. incognita* genome; hence, it was not possible to study its expression. As previously observed with SvG effectors, the 33 DG effectors map to 57 different loci in the genome, which is consistent with the allopolyploid genome structure. A total of 29 DG effector CDS map with >90% identity and across >66% of their length from the ATG position (https://doi.org/10.15454/P5YIGX). This corresponds to 51 different loci and 30 predicted gene models in the *M. incognita* annotated reference genome. The four effectors not meeting these criteria were 5C03B (86% id), 1C05B (87.7% id), 4F05B (67.5% id), and 17H02 (57.8% cov).

Except 7A01, which was absent from all the other *Meloidogyne* genomes investigated and might represent an *M. incognita*-specific gene, all of the rest of the DG effectors mapped from their ATG position in at least one other clade I *Meloidogyne* species. Although 28B04 was not found in the *M. incognita* genome assembly, this gene mapped to the genome of *M. javanica*, suggesting it might be missing in the *M. incognita* assembly. Overall, 27, 27, and 28 DG effector genes mapped from the ATG position with >90% identity and across >66% of their length in the genomes of *M. arenaria*, *M. javanica,* and *M. enterolobii*, respectively, and the majority mapped to multiple loci, consistent with their polyploid genome structures. Reconstruction of ancestral states based on the DG effector genes’ phylogenetic distribution suggests that at least ~82% (28/34) were inherited from a common ancestor of clade I *Meloidogyne* species (Figure 2). In *M. hapla* only eight DG effectors genes map to the genome and only one (14-3-3) passed the 90% identity threshold, which is consistent with the higher evolutionary distance. Thus, only ~23% of the known *M. incognita* DG effectors could be traced back to a common ancestor of clade I and clade II *Meloidogyne* species (Figure 2 and https://doi.org/10.15454/P5YIGX).

### 3.3. Hierarchical Clustering Analysis Highlights Changes in Expression of Genes Related to Parasitism as a Whole

To provide a more holistic view of gene expression, we clustered and classified the 12,461 differentially expressed genes based on their expression profiles in all life stages and studied the enriched functions and the distribution of effector genes in these clusters. Eight distinct clusters (named A to H hereafter) were generated as indicated by the C-index and measured relative to the median expression level throughout the entire life cycle (https://doi.org/10.15454/2XJCJQ). The eight clusters varied in the number of genes across an order of magnitude (from 611 to 5441 genes, Figure 3). Three distinct and sequential gene expression patterns were identified according to the timing in developmental stages of *M. incognita*. We noted that clusters A and B contained 17% of all DEGs and exhibited a peak of expression at the egg stage. These genes were enriched in GO terms related to embryogenesis, such as cuticle formation, cell adhesion, and migration (https://doi.org/10.15454/V3SCRC). We also identified an enrichment of hydrolytic enzymes (i.e., metalloendopeptidase activity) and pattern-receptors (i.e., scavenger receptor activity) which may also act in physiological and pathological processes. Interestingly, the majority of DEGs (6934) were grouped into just two clusters (C and D) characterized by a peak of expression at the ppJ2 stage. These two clusters were significantly enriched in genes associated with pathogenesis (https://doi.org/10.15454/V3SCRC).

Interestingly, cluster C was overrepresented in DEGs encoding a mixture of potential secreted plant cell wall degrading enzymes, including CAZymes (glycoside hydrolases, pectate lyases) and phosphatases (serine/threonine protein phosphatases), which are known to be involved in host colonization processes but also in survival [42,43]. Consistent with the secretion of many enzymes and other proteins, the only enriched GO term in the cellular component ontology was ‘extracellular region’. We also noted a significant enrichment of terms related to transcription factor activity in cluster C that might be related to the genome-wide response to plant invasion. In cluster D, various different GO terms related to ion channel and transport activities or DNA replication and binding were significantly enriched, probably reflecting the preparation for metamorphosis from the late J2 to J3 stage.

The next two clusters, E and F, only encompass 695 and 611 DEGs, respectively, and seem to be characterized by a peak of expression in eggs, a trough in the ppJ2, and a recovery of expression in the mix of sedentary J3−J4 larvae stage. Enriched terms at clusters E and F correspond to DNA replication, DNA binding, RNA translation/protein biosynthesis, and folding activities, probably reflecting the profound morphological remodeling and physiological changes at these stages (https://doi.org/10.15454/V3SCRC). As the third key time point in the nematode’s life cycle, the clusters G and H encompassed ~17% of DEGs and were characterized by a marked upregulation during the sedentary J3/J4 larval stages. Both clusters showed an enrichment of gene categories related to nutrient processing/energy acquisition (metabolism of carbohydrates and lipids), and detoxification process (oxidation–reduction processes). Intriguingly, the clusters G and H differed by their content of GO terms related to hydrolytic and detoxification enzymes according to the type of regulation and substrates. For example, glutathione hydrolase and serine-type carboxypeptidase were prevalent in cluster G, while metalloendopeptidase inhibitor activity was identified in cluster H. Overall, these results are consistent with the idea that there is a shift in the expression of a suite of genes at this given developmental stage driving the preparation and progression in the parasitism of *M. incognita*.

While the hierarchical clustering revealed global transcriptional changes during the time course of the *M. incognita* life cycle, we also more specifically focused on the expression patterns of effector genes over time. The 30 and 62 known *M. incognita* DG and SvG effectors mapped to 94 different genome loci in which a gene model was predicted in the genome, enabling study of their expression patterns. Overall, 22 (73.3%) DG and 52 (81.3%) SvG effector gene models showed significantly different gene expression between at least two different life stages during the *M. incognita* parasitic life cycle. Using the X^2^ test, neither the DG nor the SvG genes were randomly distributed in the eight expression clusters (*p*-value < 2.2 10^−16^). We found that cluster C was significantly enriched in known effector genes (adjusted FDR: 1.91 10^−7^). Cluster C presents a pattern of higher expression in the ppJ2 stage. SvG genes are mostly expressed (approximately 71%) at the early stage of infection, represented by the clusters C, D, and E, and positively upregulated at 65% during the pre-parasitic J2 stage. In contrast, only 1.9% of differentially expressed SvG genes were found in the clusters A and B, characterized by a peak of expression at the egg stage. The rest of the differentially expressed SvG effector genes (27%) were grouped in the last two clusters, G (21.2%) and H (5.8%). These clusters are characterized by a peak of expression at the J3/J4 stages.

Interestingly, and consistent with the previous description, differentially expressed DG effector genes were mostly (~59%) distributed in the G and H clusters. These DG effector genes were significantly upregulated (64%) at the J3/J4 stage. SvG and DG effector genes show a pattern of expression that was positively and temporally correlated with the infection course, as previously described in [44]. In the current observations, we noted that SvG and DG effectors were significantly enriched in clusters C, G, and H, which contain the core parasitism genes, including specific members of gene families encoding plant cell wall degrading enzymes and detoxification enzymes known to be involved in the parasite invasion process.

### 3.4. A Promoter Motif Is Associated with Dorsal Gland Effectors and Secreted Proteins

An emerging theme in plant-parasitic nematodes is the in silico discovery of effector regulatory motifs in several phylogenetically distant species, including the pine-wilt *B. xylophilus*, the cyst nematode *Globodera pallida,* and the root lesion nematode *Pratylenchus penetrans* [11,14,45]. These studies provide powerful insights into the understanding of the adaptation to plant parasitism in nematodes. Given that effectors in *M. incognita* are specifically regulated in space and time, similarly to other phytoparasites, we extended the search of cis-regulatory sequence motifs enriched in the promoters of genes related to RKN parasitism. From the coordinated loci corresponding to the comprehensive set of known DG and SvG effectors mapped to the *M. incognita* genome, a series of putative promoter regions were extracted (500, 1000, and 1500 bp 5′ of the start codon) using custom Python3 software (https://github.com/peterthorpe5/intergenic_regions). The 5′ regions of DG effectors and SvG effectors were compared to one another, as well as to a set of 5′ regions from 167 genes not annotated as effectors but widely conserved in nematodes other than *Meloidogyne* species (i.e., likely non-effectors). Gene predictions at the ends of scaffolds were omitted due to possibly truncated data. After this filtering step, 92% of all annotated protein-coding genes (40,312/43,718) were kept, which encompassed 90% of mapped DG CDS (46/51), and 83% of mapped SvG CDS (62/74).

Comparing DG effector gene upstream regions with putative non-effector gene upstream regions using a differential motif discovery algorithm (HOMER, [32]) identified a series of similar motifs (of approximate consensus TGCACTT) in all three length categories (500 bp: TGCACT (32% on target, i.e., of the positive set that contain the motif); 1000 bp: TGCACTT (32% on target); and 1500 bp: TTGCACTT (50% on target); Figure 4A). A similar motif was enriched in the DG effector gene upstream regions when compared to the upstream regions of SvG effector genes, suggesting that this motif is specifically enriched in regulatory regions of DG effector genes. In contrast, no consistent motifs were enriched in the set of SvG upstream regions compared to either DG or putative non-effector upstream regions.

On a whole-genome scale, 17.3% of all genes that contain one or more copies of the TGCACTT motif in their 1000 bp upstream region encode putatively secreted proteins. The proportion of putatively secreted proteins rises to over a quarter (26.9%) for genes with two or more copies of the motif, in their 1000 bp upstream region. This represents an approximately four-fold enrichment over the remainder of the genome (c.f. 6.4% putatively secreted proteins, FDR: 1.8 10^−26^). To determine whether variants of this motif provide similar enrichment, or enhance the overall enrichment, a custom Python script (https://github.com/sebastianevda/Mel_DOG_scripts) was used to search for all single base-pair mismatch variants of the motif in the 1000 bp upstream of all genes, and calculate the proportion of putative secreted proteins. Two variants were consistently enriched for putatively secreted proteins (TGCACTT and TGCCCTT, Figure 4B). We next searched in the 1000 bp proximal promoter sequences of the known DG effector gene loci to further explore the occurrence of the two variants. The distribution of the variants revealed a clear preference for one variant over the other in the upstream regions of known effectors, comprising 43.5% with only TGCACTT (20/46), while 3/46 had only TGCCCTT, and 4/46 had both TGCACTT and TGCCCTT. On a genome wide-scale, 20% of all genes that encode a putative secreted protein have one or more copies of TGCMCTT in the 1000 bp 5′ region (524/2737); nearly half (48%; 251/524) have the variant TGCACTT, followed by 37% (194/524) with TGCCCTT, and 15% (79/524) with both variants. For a random set of genes, 4% (3/74) have TGCACTT, 4% (3/74) have TGCCCTT, and 1.3% have both (1/74). Consequently, TGCMCTT is enriched approximately 10-fold over random in the promoters of genes encoding putatively secreted proteins, and hereafter considered as a core motif for all further analyses.

Generally, the more copies of the motif in the upstream region of a gene, the more likely the corresponding gene encodes a putative secreted protein (Figure 4B). A consistent and positive correlation was observed between the proportion of putatively secreted proteins and the number of copies of the motif in the upstream region, at all lengths categories from 300 to 10,000 bp (Figure 5A). The number of DG effector genes identified using 10,000 bp upstream regions was used to set an upper limit for the DGs that are identifiable using this motif (60.87% (28/46)). Of these motif-identifiable DG effector genes, 95% were recovered by searching for one or more motifs within the first 1000 bp upstream region (Figure 5B). To determine whether the predictive power of this motif is restricted to the non-coding upstream region of a gene, the enrichment analysis was repeated to include 200 bp of coding sequence. The inclusion of 200 bp of coding sequence reduced the proportion of genes encoding putatively secreted proteins, and is therefore detrimental to prediction (e.g., Figure 5C). This supports the motif as a putative cis-regulatory element.

Finally, we determined the positional enrichment of the motif within the upstream regions of DG effectors, and of all genes that encode putatively secreted proteins. Plotting the frequency distribution of TGCMCTT in known DG effectors from 200 bp downstream of the start codon to 1000 bp upstream reveals a peak between −350 to −150 bp (Figure 5D). The location of this peak is consistent with both the lack of enrichment for secretory proteins using 100 bp upstream of the start codon (Figure 5A), and the worse enrichment when including coding sequences (Figure 5C), because the peak is non-coding and occurs further upstream. For those genes that contain at least one copy of the motif in the 1000 bp upstream of the start codon and encode a putative secretory protein but are not already classified as a known DG effectors, a similar frequency distribution of the motif is found (Figure 5E), complementary to the known DG set, with peaks at approximately −200 and −500 bp. Combining these distributions provides a broad peak from −150 to −500 bp for motif-identifiable putatively secreted proteins.

Taken together, these data describe a possible cis-regulatory promoter motif present in the first 1000 bp upstream of the majority of previously known DG effector loci (approximately 60%) and highlights a substantial but reasonable subset (approximately 20%) of the genes that encode putatively secreted proteins that have not yet been characterized as actual DG effectors. Moreover, motif-identifiable putatively secreted proteins are also enriched in the differential expression cluster C (peak at J2, FDR = 1.91 10^−7^), consistent with a role in parasitism. Similar to previous dorsal gland cell motifs in other plant-parasitic nematodes, this motif may provide a way to predict root-knot nematode DG effectors, narrow down the candidate effector repertoire, and by extension explore the regulation of effectors in root-knot nematodes.

To arrive at a refined list of putative DG effector proteins for future studies, we combined the predictive power of a variety of attributes, including: (i) the presence of a signal peptide for secretion; (ii) the absence of transmembrane domain; and (iii) the presence of one or more cis-regulatory motifs within the first 1000 bp upstream of the start codon. A total of 457 candidates fulfilled these criteria. Of note, 77% of these candidates also had a predicted MERCI motif in the encoded protein sequence [18,34]. Putative DG effector candidates identified in this way were assigned GO terms based on their InterPro domain annotation. This revealed a significant enrichment in two biological processes (proteolysis (GO:0006508) and carbohydrate metabolic process (GO:0005975)), two molecular functions (hydrolase activity/hydrolysing O-glycosyl compounds (GO:0004553) and cysteine-type peptidase activity (GO:0008234)), and two cellular components (extracellular region (GO:0005576) and acetylcholine-gated channel complex (GO:0005892); Figure 6). Interestingly, and consistent with previous descriptions, most novel predicted DG effector genes were found in clusters C and G, respectively characterized by an upregulation at the ppJ2 and J3/J4 stages. Despite only ~26% of putative DG effectors (118/457) are functionally annotated with at least one GO term, these data are consistent with what is known about the functions of previously characterized effectors. The list of 457 predicted DG effectors, their functional annotation, and the full results of enriched GO terms are available at https://doi.org/10.15454/2O77EF.

Finally, to broaden the utility of this motif to identify novel putative effectors, the DG promoter enrichment analyses were repeated with four other species across the genus (Figure 7). Extremely similar motifs were identified as significantly enriched in 2 kb promoter regions of *M. incognita* DG effectors mapped to the *M. arenaria, M. javanica*, and *M. enterolobii* genomes when compared to a control set of non-effector promoters in each respective genome. The analysis could not be reliably reported for the promoters of *M. hapla* DG effectors because the number of *M. incognita* DG effectors unambiguously mapping to the genome (8) is too low to draw meaningful conclusions. Taken together, we conclude that this promoter motif (named Mel-DOG for ‘Meloidogyne DOrsal Gland’), and thereby its utility to predict effectors, is broadly conserved across at least the clade I root-knot nematodes.

## 4. Discussion

We performed a genome-wide survey of *M. incognita* transcription to identify ‘trends in’ and the causes of gene expression that likely contribute to nematode parasitism. These analyses enabled us to identify biological signatures of key developmental transitions during the lifecycle. We approached the question of parasitism gene regulation in a spatio-temporal manner. Together, these results provide a step forward in our understanding of root-knot nematode transcriptional regulation by the identification of a putative cis-regulatory element, Mel-DOG, associated with DG effector gene expression in *Meloidogyne* spp. By extension, these discoveries reveal an additional criterion on which to predict DG effectors in the most economically important plant-parasitic nematode genus.

Despite an increased amount of information regarding the mechanistic aspects of plant–nematode interactions, there is a paucity of molecular information about the parasite itself, i.e., the life stage specific gene expression in *M. incognita,* with a particular focus on parasitism genes [46,47]. We analyzed the transcriptome time-course of the RKN comprising four crucial stages of development: two pre-parasitic stages (eggs and ppJ2) and two parasitic stages (a mixture of J3/J4 larvae and adult females). Our data highlight gene functions that are consistent with the abilities of the nematode to complete critical transitions and ultimately become an enduring and successful parasite [4]. We thus followed the nematode’s lifecycle from migratory to sedentary and describe the key functional classes of the genes employed.

At early stages (transition from eggs to ppJ2), *M. incognita* expresses genes involved in chemosensation and movement. These observations make sense with the particular time point of the pre-parasitic life stages of the animal, in which the triggering of signaling pathways accompanies hatched ppJ2s that actively sense root exudates and move towards their host, supporting host recognition and attraction. The induction of these groups of genes correlated with the preparation of parasitism and have been reported in other PPNs, including *Meloidogyne graminicola* [48], *Globodera pallida* [49], and the migratory endoparasite *Radopholus similis* [50]. The RNAi mediated downregulation of such chemosensory (e.g., *odr* genes) or neuropeptide genes (e.g., *npl-3* or *npl-12* genes) resulted in a significant alteration in the attraction and penetration of *M. incognita* ppJ2 to host roots, supporting their importance at early events of plant parasitism [51,52]. Similarly, we observed the upregulation of genes responsible for carbohydrate degradation as a signature of this transition. It is known that the motile J2 nematode produces and secretes a myriad of enzymes for plant cell wall softening/degradation during the invasion of root tissues [38,53,54]. Silencing certain cell wall degrading enzymes reduced root invasion by *M. incognita* [55,56] and other PPN species, such as the migratory endoparasite *Pratylenchus coffeae* [57] and *Globodera rostochiensis* [58].

A second transcriptional step was observed from the early migratory to sedentary phases (ppJ2 to J3/J4). During this transition, the juvenile undergoes physiological and developmental changes, highlighting that this transition is a turning point for the parasite. The downregulation of genes involved in chemosensory behavior suggests that once inside the root, the parasite no longer needs to follow attractive stimuli. The success of infection mostly relies on its ability to cope with the first line of plant defense and immunity and to move towards the host tissues [49,59,60,61]. At the J3/J4 stage, significantly upregulated genes included putatively secreted proteins and enzymes such as superoxide dismutase and serine-type carboxypeptidase that may be involved in stress and defense responses. We hypothesize that they may allow the protection of the endoparasite from the toxic effects of damaging ROS produced by the surrounding host plant cells, as supported by the RNAi-mediated downregulation of peroxiredoxins (PRXs) in *M. incognita* negatively impacting on its parasitic ability [62]. At this stage of infection, the juvenile probably does not acquire nutrient or energy sources from the plant host, but may instead utilize its lipid reserves [46,47,52] and also undergoes morphological changes (molting and reabsorption of the cuticle). Consistent with this, significantly enriched GO terms related to lipid metabolism were identified.

In comparison with the previous developmental transition, the transition from J3/J4 to adult female is marked by a return to feeding and the production of hundreds of eggs. A particularity of this transition was the downregulation of genes associated with locomotion, supporting the idea that the parasitic worm does not need to move once the feeding site is established. Conversely, we observed an upregulation of genes predicted to encode hydrolytic enzymes (i.e., serine-type endopeptidase) at this transition. While some of these genes may be involved in the evasion or suppression of plant immune responses [43,55,63], their roles are manifold and require functional characterization to better understand their importance in nematode biology, as well as in pathogenicity.

Finally, to complete the *M. incognita* lifecycle, a comparison of the egg to the adult female stages showed a diverse array of gene expression changes necessary for embryogenesis which, in a sense, marks the achievement of nematode parasitism with the production of offspring. As reported in a previous study, our data analyses also suggest the differential regulation of genes involved in various basic biological processes (e.g., carbohydrate, amino acid, and lipid metabolism [64], the semaphorin/plexin pathway, and cell adhesion and cuticle formation). At this step of the lifecycle, a dampening of expression of genes related to stress responses occurs (e.g., superoxide dismutase activity, oxidoreductase activity, L-ascorbic acid binding), suggesting that the host is no longer a threat for the parasitic animal, or the threat is no longer of concern since the adult female basically dies while extruding eggs in a gelatinous protective matrix on the root surface.

In addition to the DEG analysis at the transition between the four life stages, we also performed hierarchical clustering of DEGs to define and study groups of genes that share similar expression patterns during the parasite life cycle. DEG clusters showing peaks of expression at the pre-parasitic J2 and parasitic J3/J4 stages were associated with genes encoding for members of the *M. incognita* degradome [63]. Genes encoding a subset of enzymes, hydrolases (GO:0004553), pectate lyases (GO:0030570) and protein tyrosine/serine/threonine phosphatases (GO:0008138) were overexpressed at ppJ2, while the subset of glutathione hydrolases (GO:003674) and serine-type carboxypeptidases (GO:0004185) showed a peak of expression at the J3/J4 stages. In agreement with previous reports, our data suggest that genes encoding metalloendopeptidases are likely to be involved in early events of *M. incognita’*s life cycle, with a significant upregulation at the egg stage (cluster A) and, conversely, a downregulation at the J3/J4 stages (cluster G). Metallopeptidases are known to be mainly expressed at the egg and ppJ2 stages in *M. incognita* [54], while others, such as cysteine peptidase, are highly expressed at later stages. Both subsets of proteases are largely represented in the *M. incognita* genome [18,63,65,66]. The paramount importance of these enzymes has been proposed previously in nematode parasitism; for example, a transmembrane metalloprotease, Neprilysin, is positively differentially expressed in *Heterodera avenae* during exposure to root exudates [67] and may be involved in the regulation of pathogenicity [68]. Known to be physiologically essential for many organisms, including PPNs, but globally restricted to Metazoans, they are referred to as core parasitism genes [69] because they are associated with a role in tissue maceration/penetration and common to both parasitic nematodes of plants and animals [70,71,72,73]. In addition, these transcriptomic profiles associated with GO term analysis described in *M. incognita* mirrored similar studies in other PPNs, supporting the general commonalities that define the parasitic lifecycle [63,70,74], but the proportion and type of these enzymes vary according to the lifecycle [11,18,50]. Peptidases and CAZymes are both the result of a convergent adaptation in parasitic nematodes and an apparent ancient microbial origin, acquired by multiple independent lateral gene transfers from different bacterial and fungal sources [37,38].

Like many other PPN species, most *M. incognita* known effectors are produced from the two main sets of secretory gland cells, DG and SvG, for which secretions are regulated at a specific stage during the lifecycle of the nematode. Previous works showed that SvG are more active during nematode penetration, while DG is mostly involved in the formation and maintenance of nematode feeding [75,76,77]. For the root-knot nematodes, it is not yet clear how evolutionarily conserved known effectors are across the genus, nor how the expression of effectors is distributed across the lifecycle of a species. To investigate the evolutionary conservation of known *M. incognita* SvG and DG effectors, we mapped the corresponding CDS sequences to the genomes of *M. incognita* [16] and four other *Meloidogyne* species. Three species (*M. arenaria*, *M. javanica* and *M. enterolobii* [16,29]) were from the same RKN clade I [41], while *M. hapla* [30] was from clade II and thus more distantly related. More than 87% (42/48) and 82% (28/34) of the known *M. incognita* SvG and DG effectors, respectively, could be traced back as inherited from a common ancestor of all the clade I *Meloidogyne* investigated here. Alignment of effector CDS to the clade II *M. hapla* genome allowed us to trace back an even more ancient origin in a common ancestor of clade I and II *Meloidogyne* for 52% (25/48) of SvG effectors, but only about 23% (8/34) of DG effectors were known in *M. incognita*. Only a more comprehensive sampling of *Meloidogyne* species in the different clades for genome sequencing will allow us to refine their precise evolutionary origin and path in the future.

The generally lower conservation of DG effectors compared to SvG effectors has several plausible explanations that are not necessarily mutually exclusive. Firstly, SvG effectors are generally deployed at the earlier stages of parasitism, are involved in migration through host tissue, and are conserved within the migratory endoparasites (i.e., Pratylenchus) that predate the genus [15]. These effectors are thus probably involved in core parasitic functions common to multiple plant-parasitic nematodes and have probably evolved under purifying selection limiting divergence at the sequence level over evolutionary time. The DG effectors, in contrast, are generally accepted to be responsible for the precise manipulation of host physiology and development. They are therefore likely more recent, and under positive selection imposed by the different defense systems of the host. This might explain their lower degree of sequence-level conservation in the *Meloidogyne* genus and lack of identifiable orthologs in other species.

We subsequently cross referenced this comprehensive known effector catalogue with gene expression data to describe patterns of transcription throughout the lifecycle of *M. incognita*. This revealed that two main waves of effector expression in pre-parasitic and parasitic stages occur. Genes encoding known SvG effectors were mostly differentially expressed (81.3%) and upregulated during the ppJ2 stage (clusters C and D), while 73.3% of DE DG effector genes were associated with the sedentary J3/J4 stage (clusters G and H), most of which were upregulated at this specific time point. Similar transcriptome analyses of the cyst nematode *Globodera pallida* showed three waves of effector expression (pre-parasitic (J2), early (7/14 dpi), and late (21−35 dpi)), with a fourth category expressed throughout the biotrophic stages [11]. We observed that DG and SvG gene expression patterns coincide with those of certain subsets of degrading/modifying enzymes and those of stress responses which, intuitively, may act in concert to orchestrate the successful parasitic process [9]. In a previous study [56], the transcriptional change in another set of CDWE effectors has been demonstrated through the RNAi silencing of pioneer MSP effectors, supporting a link between the expression of pioneer effectors and activities of such CDWEs in the RKN *M. incognita*. Overall, our analysis of the global gene expression coupled with GO annotation gives a more detailed view on the deployment of these effectors over time.

In addition to temporal control, the transcription of effectors is also precisely spatially regulated in the DG and SvGs. Recent exciting discoveries of cis-regulatory elements in PPNs associated with spatial regulation strengthen the interest to identify a putative regulatory promoter motif in the RKN *M. incognita* [11,14,45]. Using a combination of in silico and experimental transcriptomic analyses, we have thus identified a putative non-coding DNA motif, TGCMCTT (named Mel-DOG), specifically enriched in the promoter regions of effector genes experimentally validated for dorsal expression in *M. incognita* for the first time (references in https://doi.org/10.15454/P5YIGX). Mel-DOG is conceptually similar, but unrelated sequence-wise, to other cis-regulatory sequence motifs among PPNs. Cis-regulatory sequence motifs associated with gland cells do not appear to be conserved between the major lineages. For example, the DOG box, first identified in *Globodera* spp., is descriptive and predictive of effectors in cyst nematodes but not RKN nor pine wilt nematodes [11,13,14,15], and the STATAWAARS motif first identified in *B. xylophilus* is descriptive and predictive of effectors in this species, but not cyst nematodes nor RKN [11,14,45]. Similarly, Mel-DOG appears to be associated with effectors in the RKN but not the cyst nor the pine wilt. Finally, Mel-DOG follows the trend that no motifs specifically associated with the SvG gland cells have been identified so far.

Following the identification of Mel-DOG, we used this motif as an additional criterion to predict effectors *in silico*. Consistently, Mel-DOG enrichment in the RKN genome was identified in promoters of genes encoding secreted proteins and comprising certain specific functions in pathogenicity, such as terms related to CAZymes. For instance, a-L-Fucosidase (GH29) is included in the top 20 most represented CAZyme families from the genome-wide analysis of excretory/secretory proteins in *M. incognita* [54] and from the secretome of the pine wilt nematode *B. xylophilus* [78,79]. Using hierarchical clustering, Mel-DOG was functionally enriched within DEGs from cluster C, the main cluster comprising a major transcriptional shift in gene expression with a sharp peak at ppJ2, and although not statistically supported, Mel-DOG is also likely associated at a late stage of development (larval J3/J4; cluster H). These data support the involvement of Mel-DOG in the transcriptional regulation of degrading/modifying enzymes, pointing to a role in tissue maceration/penetration and a specific temporal window of Mel-DOG gene expression during *M. incognita*’s lifecycle. Of importance, Mel-DOG is also associated with genes encoding putative secreted proteins with unknown function, in keeping with previous studies on other PPNs. Overall, the existence of non-coding DNA motifs associated with effector genes adds an important piece to the puzzle of parasitism gene regulation in PPNs. It remains undoubtedly essential to better understand the Mel-DOG function at a specific stage to further unveil its regulatory role during nematode parasitism and to extend this discovery by dissecting gene regulation networks that link with promoter motifs. With respect to insights, one ongoing hypothesis is that a small number of regulators act in concert for effector expression, and thus, the gene disruption of these few regulators can simultaneously disrupt the function of hundreds of putative effectors and have a major effect on parasitism. Progress in understanding effector biogenesis and biology is limited by a lack of tools for functional genetics on plant-parasitic nematodes, mostly due to their biology and gonad accessibility, but although this is challenging, recent improvements in mRNA delivery appear realistic and promising [80]. The current approaches for their study are mostly based on the analyses of a single effector at a time through its overexpression in plants, target gene silencing via RNAi, and the identification of target host proteins [9]. However, silencing of a single effector gene can be difficult to interpret because RNAi is never done at 100% efficiency, sometimes it can have a redundant role, and by consequence, its functional role may be not enough to completely alter nematode parasitism [9,81,82,83]. Thus, targeting the regulators associated with Mel-DOG might represent an attractive and accurate target to develop novel nematode control strategies. For example, a well conserved Zn2Cy6 transcription factor (PnPf2) is involved in the expression of genes associated with nutrient assimilation and numerous effectors in the phyto-necrotrophic fungus *Parastagonospora nodorum*. Disrupting the transcription factor expression dramatically alters the plant cell wall degradation, asexual reproduction, and virulence [84].

In addition to the potential for translation, the discovery of Mel-DOG provides additional criterion to predict and annotate effector repertoires in *Meloidogyne* genomes. Prior to this study, the prediction of nematode effectors was based on the identification of a putative N-terminal secretory peptide, no transmembrane region, and/or MERCI motif in their 100 first amino acids [18,34]. Thus, the promoter motif complements the roadmap for the discovery of PPN effectors in general and, for the first time, it might be extended in the RKN *M. incognita* [85,86]. Here, we combine our understanding of effectors to provide a high-confidence list of putative effectors for future validation and study.

## 5. Conclusions

The life stage specific transcriptome of *M. incognita* provides a global overview of biological functions, and, by extension, may highlight the challenges associated with parasitism. We identify both a core set of effectors, and an associated promoter motif, that are broadly conserved across clade I *Meloidogyne* spp. The identification of the Mel-DOG box motif opens up a new way to study effector biogenesis in root-knot nematodes and provides an additional important criterion for the in silico prediction of effectors.

## Figures and Tables

**Figure 1 genes-12-00771-f001:**
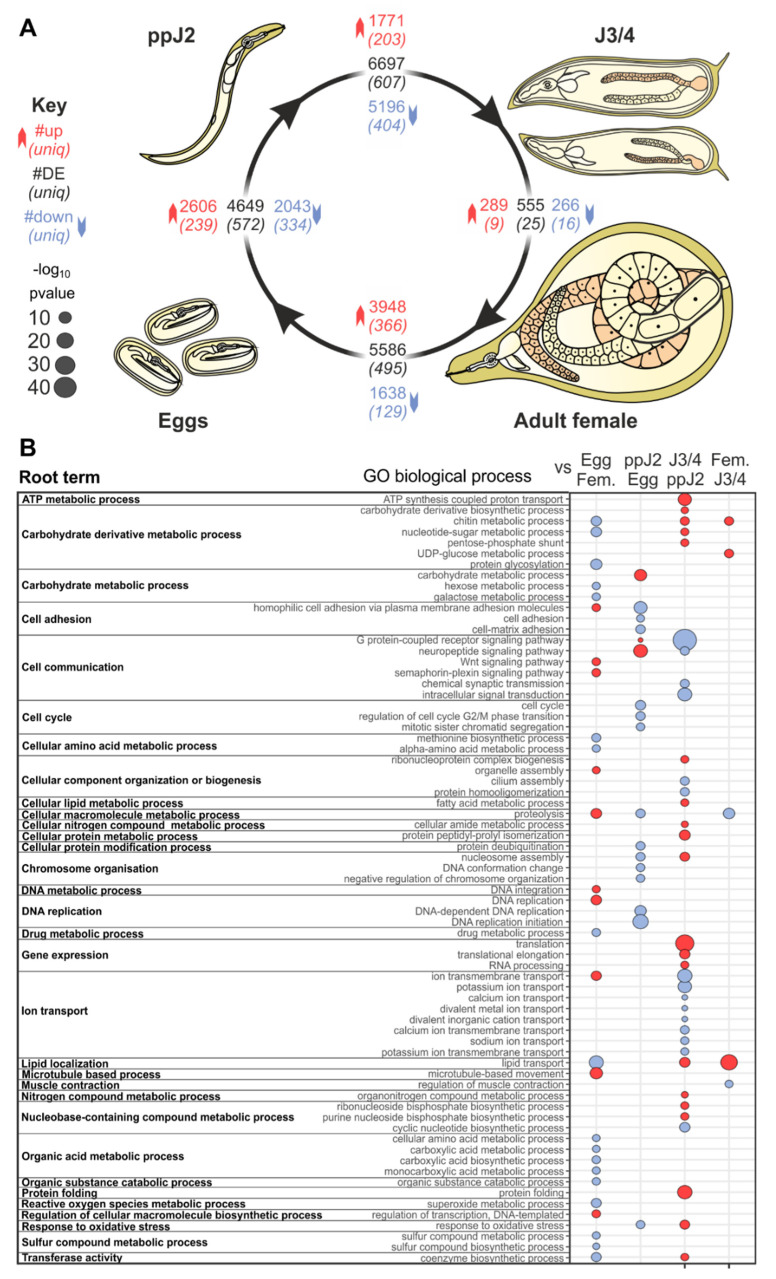
Distinct patterns of 12,461 differentially expressed genes (DEGs) and gene ontology (GO) term enrichment between the life stages of *M. incognita.* (**A**) Representation of the nematode life cycle. At each transition between stages (black arrows), the total number of DEGs is indicated in black in the central box. Genes significantly upregulated and downregulated are indicated in red and in blue, respectively. The number of DEGs only at this transition is indicated in italics and parentheses. (**B**) The GO biological process terms significantly enriched in each transition are shown, grouped by their root terms. Bubble plots illustrate the most significant GO terms that were overrepresented (in red) and underrepresented (in blue) for each term. If no bubble is present, the representation of this term was not significantly different at that transition. Bubble size is calculated as −log10 (refined *p*-value), resulting in a *p*-value FWER threshold at 0.0125.

**Figure 2 genes-12-00771-f002:**
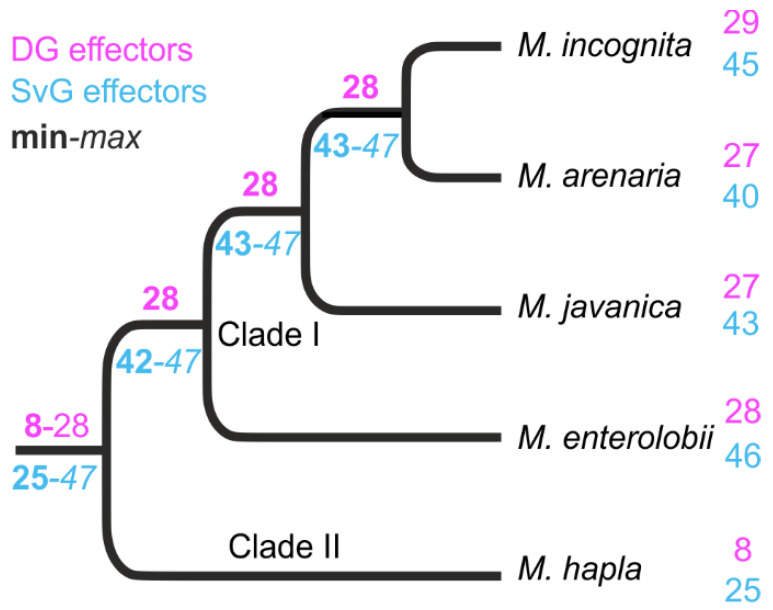
Reconstruction of ancestral numbers of homologues of *M. incognita* effectors across the *Meloidogyne* phylogeny. Based on the mapping of known *M. incognita* SvG (cyan) and DG (magenta) effectors on the genomes of five *Meloidogyne* species, ancestral numbers were reconstructed using parsimony. Minimal effector numbers are given in bold at the corresponding branches and maximal numbers in italics, depending on whether the equi-parsimonious absence/presence of effectors are considered actually absent/present.

**Figure 3 genes-12-00771-f003:**
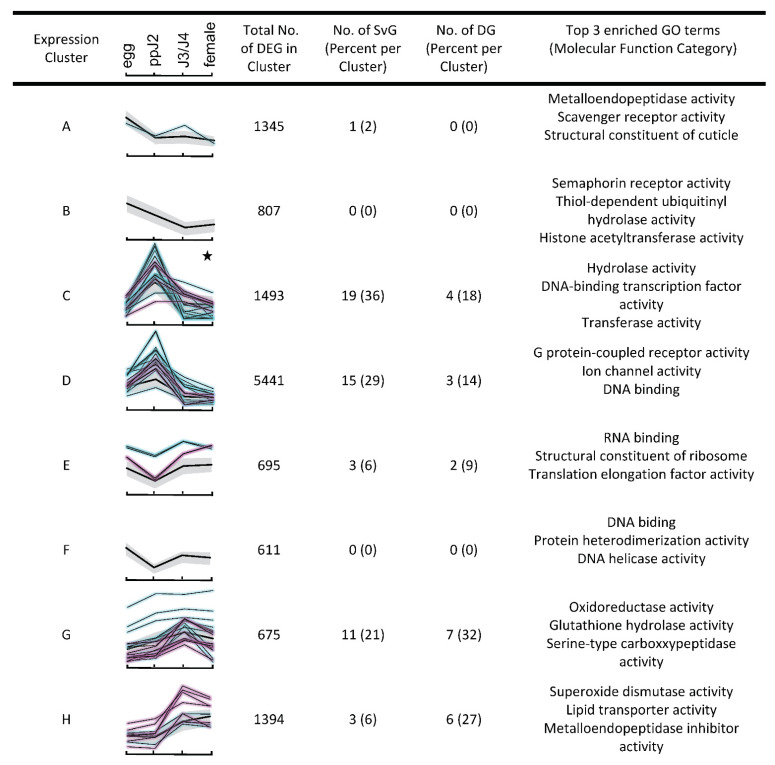
Temporal gene expression profiles throughout *M. incognita* infection. Hierarchical clustering identified 8 distinct temporal patterns of differentially expressed genes and known effector genes (SvG and DG effectors in cyan and magenta lines, respectively). The *x* axis represents four developmental stages. Black stars indicate clusters significantly enriched in the cellular component term ‘extracellular region’. Black lines represent the average profile for each cluster.

**Figure 4 genes-12-00771-f004:**
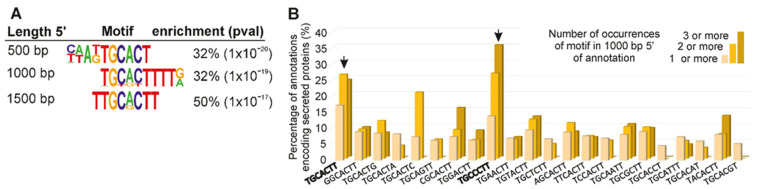
Identification of a promoter motif associated with DG effector genes in *M. incognita*. (**A**) Selected similar motifs enriched in 500, 1000, or 1500 bp promoter regions of *M. incognita.* DG effector genes compared to a set of non-effectors. A core motif derived from these predictions is TGCACTT. (**B**) The proportion of annotations that encode secreted proteins is shown for those that contain one (yellow), two (orange), or three (dark orange) copies of each 1 bp mismatch variant of TGCACTT.

**Figure 5 genes-12-00771-f005:**
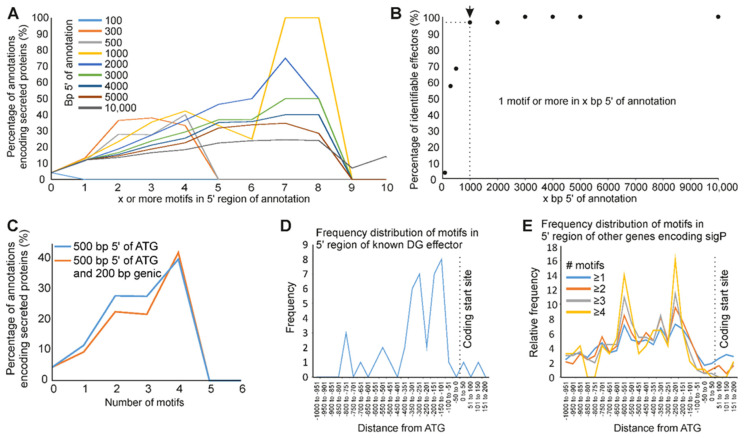
Location of a promoter motif associated with dorsal gland effector genes in *M. incognita*. (**A**) For promoters that contain x or more copies of the motif (where x varies from 0 to 10), the proportion of annotations that encode putatively secreted proteins is shown. Colors represent repeats of the analysis with different promoter lengths (ranging from 100 bp to 10,000 bp). (**B**) For effectors that contain one or more motifs in the 10,000 bp upstream of the coding start site, the proportion that are identified with an x bp promoter (where x varies from 100 to 10,000) is shown. (**C**) Using either 500 bp upstream of the ATG (blue), or 500 bp upstream and 200 bp downstream of the ATG (orange), the relationship between the number of promoter motifs and whether the corresponding genes encode putatively secreted proteins is shown. (**D**) Frequency distribution of motifs in the promoters of known DG effector genes. (**E**) Frequency distribution of motifs in the promoters of other genes (i.e., excluding known effectors).

**Figure 6 genes-12-00771-f006:**
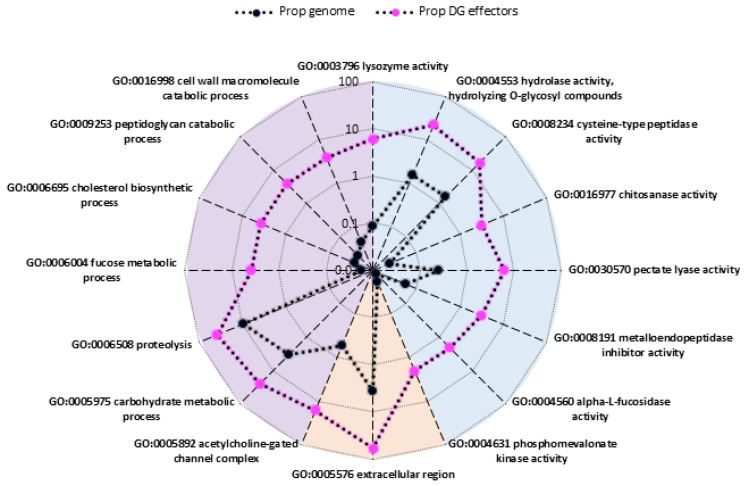
Radar plot with the GO terms overrepresented in the candidate DG effectors relative to the whole genome in a logarithmic scale. Dotted curves represent the percentage of proteins with a GO term having this specific GO term assigned (black: whole genome; magenta: candidate DG effectors). Biological process, molecular function, and cellular component GO terms are indicated with a purple, blue, and red background, respectively.

**Figure 7 genes-12-00771-f007:**
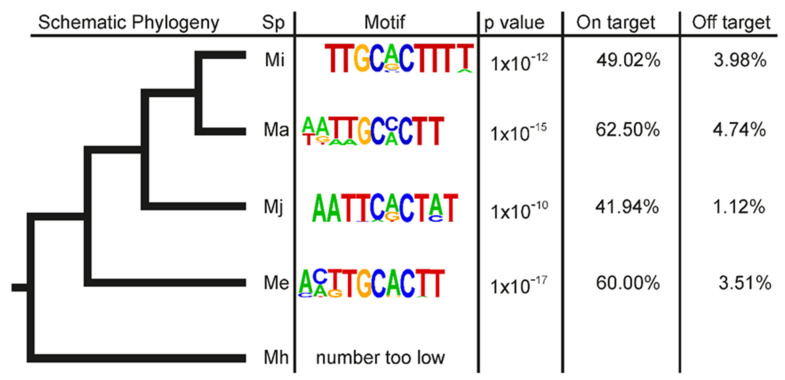
Independent identification of DG promoter motifs in the Meloidogyne genus. Left, a schematic representation of the phylogenetic relationship between *M incognita* (Mi), *M. arenaria* (Ma)*, M. javanica* (Mj), *M. enterolobii* (Me), and *M. hapla* (Mh). Right, selected motifs identified as enriched in the promoter regions of DG effector genes when compared to a set of non-effectors. On target represents the proportion of DG effector genes that possess this motif in their 2 kb upstream region. Off target represents the proportion of non-effectors that contain this motif in their 2 kb upstream region.

## Data Availability

All data supporting the results presented in this manuscript were deposited and are publicly available at the INRAE institutional data verse at the following URL: https://data.inrae.fr/dataverse/transcriptome-MincV3, with each dataset referred to at the appropriate section of the manuscript using DOIs.

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
