# Peer review of "Genome Expression Dynamics Reveal the Parasitism Regulatory Landscape of the Root-Knot Nematode Meloidogyne incognita and a Promoter Motif Associated with Effector Genes"

_genes, 2021, doi:10.3390/genes12050771_

Round 1
Reviewer 1 Report
This is a well prepared article presenting new information on "Genome expression dynamics reveals parasitism regulatory landscape of the root-knot nematode Meloidogyne incognita and a promoter motif associated with effector genes" worthy of publication. This study provides excellent reports of a life stage-specific transcriptome of Meloidogyne incognita combined with an available annotated genome, and explore the spatio-temporal regulation of gene expression. In addition, reveal gene expression clusters and predicted functions that accompany the major developmental transitions. Focusing on effectors, this study identify a putative cis-regulatory motif associated with expression in the dorsal glands: providing an insight into effector regulation and combine the presence of this motif with several other criteria to predict a novel set of putative dorsal gland effectors. Also, show this motif, and thereby its utility, is broadly conserved across the Meloidogyne genus and termed it Mel-DOG. Gives excellent results taken together, and provides the first genome-wide analysis of spatio-temporal gene expression in a root-knot nematode, and identify a new set of candidate effector genes that will guide future functional analyses. These studies are worth publishing.
Author Response
We would like to thank Reviewer 1 for the positive and encouraging comments and are glad the work we presented was appreciated. No modification was asked by reviewer 1.
Reviewer 2 Report
The manuscript entitled "Genome expression dynamics reveals parasitism regulatory landscape of the root-knot nematode Meloidogyne incognita and a promoter motif associated with effector genes" shows which genes are expressed in different RKN stages, including eggs, J2, J3-J4 and females. This work is very interesting from many aspects, as no study has previously shown this detailed, life-stage specific transcriptional changes in nematodes throughout their life cycle. I believe that this research will be interesting to many and will open new questions for future research.
The Introduction is very informative and interesting, good job!
Minor observations: some parts in the Results section (e.g. Line 299-309; Line 319-321) can be moved to Discussion.
Paragraph 624-640 needs reformatting.
Author Response
We would like to thank reviewer 2 for the ensemble of positive and encouraging comments and for the relevant points that were raised and that we corrected for better clarity of the manuscript.
Concerning sections of the discussion 299-309 and Line 319-321; we fully agree with reviewer 2, and these sections of the manuscript were actually quite redundant with the discussion. Thus, they were moved and fused within the discussion where they better fit and this allowed eliminating some redundancy.
Concerning paragraph 624-640, we thank reviewer 2 for pointing us to this section of the manuscript that was unclear. We completelyreformatted and clarified this paragrapg, including the Figure 6 legend.